# Power-Efficient Recycling Folded Cascode Operational Transconductance Amplifier Based on Nested Local Feedback and Adaptive Biasing

**DOI:** 10.3390/s25082523

**Published:** 2025-04-17

**Authors:** Chunkai Wu, Peng Cai, Jinghu Li, Jin Xie, Zhicong Luo

**Affiliations:** 1School of Computer and Information Technology, Fujian Agriculture and Forestry University, Fuzhou 350002, China; chunkaiwu1105@163.com; 2Department of Information and Electromechanical Engineering, Jinshan College, Fujian Agriculture and Forestry University, Fuzhou 350002, China; 0236798063@stu.fafu.edu.cn; 3School of Mechanical and Electrical Engineering, Fujian Agriculture and Forestry University, Fuzhou 350002, China; jinxie1216@126.com (J.X.); zcl@fafu.edu.cn (Z.L.)

**Keywords:** enhanced recycling folded cascode (ERFC), high efficiency, transient response, adaptive biasing, nested local feedback technique

## Abstract

In this paper, we present a novel enhanced recycling folded cascode (ERFC) operational transconductance amplifier (OTA), which exhibits high efficiency and a fast transient response under weak inversion. Our innovative combination of adaptive biasing with nested local feedback (ABNLF) effectively enhances the input transconductance and slew rate (SR), thus improving the transient response. By incorporating coupling capacitors at the output stage, we achieve a stable operating region with large signal responses. Both the traditional RFC OTA and the proposed ERFC OTA were designed in a 0.18 μm CMOS process, operating at a power supply of 1.8 V, with quiescent currents of 8 μA and 10.4 μA, respectively. Post-layout simulations reveal a remarkable enhancement in the proposed ERFC OTA over the traditional RFC OTA, with the SR and gain–bandwidth (GBW) surging by 120- and 5.95-fold, respectively. This advancement boosts the efficiency of the traditional RFC OTA and provides an impressive figure of merit (FoM) of 130.04 (V/μs)·pF/μA.

## 1. Introduction

In analog and mixed-signal circuits, low-power operational transconductance amplifiers (OTAs) play a crucial role and are commonly used in a range of applications, such as in portable devices, Internet of Things sensors, and medical electronic devices [1,2,3,4]. The folded cascode (FC) OTA, known for its balanced gain and signal swing, is a popular choice among different OTA architectures. Designers typically favor the PMOS-input FC OTA, as it features a higher non-dominant pole, lower flicker noise, and a reduced common-mode level compared to the NMOS variant [5,6,7,8,9,10,11]. However, traditional FC OTAs may be inefficient since they rely on current sources in the folded stage, which limits their performance at lower current levels. To resolve this matter, the recycling folded cascode (RFC) OTA is introduced to enhance FC OTA performance. By substituting the folded-stage current source with active current mirrors, the RFC OTA achieves a higher gain–bandwidth (GBW) and slew rate (SR) with no change in its power consumption [12,13,14]. Nevertheless, the performance in terms of GBW, SR, and noise is influenced by the current mirror ratio, which may have a destabilizing effect by shifting non-dominant poles to lower frequencies and affecting power consumption. Moreover, the fixed tail current source restricts the slew rate.

Numerous techniques have been proposed to tackle the challenges associated with the traditional RFC OTA. These techniques include current shunt methods, double recycling FC OTAs, positive-feedback techniques, and multi-path recycling methods. An improved RFC OTA, as discussed in [15,16], effectively increases the ratio of the dynamic current source without requiring additional power consumption. In a study outlined in [17], a positive feedback mechanism implemented through cross-coupling achieves effects similar to those described in [18]. Additionally, strategies such as double RFC (DRFC) [19] and multi-path RFC (MRFC) [20] have been harnessed to enhance RFC performance. However, the intricate nature of these circuit structures introduces additional low-frequency zero-pole pairs, potentially constraining performance in high-frequency applications. One common limitation of these methodologies is the Class A operation of the amplifier. In the context of the RFC OTA, a technique known as local common-mode feedback (LCMFB) has been applied (as discussed in [21]). Nevertheless, the slew rate (SR) enhancement remains restricted due to the Class A operation of the input stage and the finite resistance associated with LCMFB. A recent development in this field is the introduction of a super Class AB RFC OTA, as proposed in [12,22], which significantly enhances both the gain–bandwidth (GBW) and SR. However, achieving high transconductance at low static current requires a large resistor, resulting in a substantial spatial footprint. Furthermore, the SR is limited due to the constant bias of the cascode transistors, as indicated in [23,24].

To address the above limitations, this work proposes an improved RFC OTA with low power consumption. A novel adaptive biasing and nested local feedback topology is introduced in the proposed ABNLF technique, enhancing the input equivalent transconductance without consuming extra power. This method resolves the trade-off between power efficiency and GBW. Additionally, a floating bias network is used for the output-stage cascode transistors, ensuring stable operating conditions for the OTA under large signal conditions. The designed ERFC OTA demonstrates good performance in terms of gain–bandwidth, power consumption, and noise.

The article is structured as follows. Section 2 provides a brief analysis of the conventional RFC OTA and introduces the principle behind the proposed ERFC OTA. Section 3 presents the circuit implementation along with a detailed analysis of low- and large-signal performance. Post-layout simulation results are discussed in Section 4, followed by the conclusion in Section 5.

## 2. Methodology of ERFC OTA

### 2.1. Limitations of Conventional RFC OTAs

The conventional RFC OTA is shown in Figure 1, assuming that ${\left(\frac{W}{L}\right)}_{1B}={\left(\frac{W}{L}\right)}_{1A}$, ${\left(\frac{W}{L}\right)}_{3B}$:${\left(\frac{W}{L}\right)}_{3A}$ = *K* and *K* is a constant. Its equivalent transconductance and gain–bandwidth (GBW) can be expressed as follows:(1)$$g_{m,RFC}=\left(1+K\right)g_{m1A}$$(2)$${GBW}_{RFC}=\frac{\left(1+K\right)g_{m1A}}{2\pi C_{L}}$$
where $g_{m1A}$ is the transconductance of $M_{1A}$, $g_{m1A}=g_{m1B}$, $C_{L}$ is the load capacitance of the OTA, and factor *K* is usually set to 3 to ensure that the RFC OTA has the same power consumption as the FC OTA.

When a large-signal $V_{id}$ is introduced to the input pair, the largest output current is limited to $2KI_{B}$. Therefore, the slew rate (SR) of the RFC OTA can be expressed as follows:(3)$$SR_{RFC}=\frac{2KI_{B}}{C_{L}}$$

Increasing *K* has a dual positive impact on the performance of both the GBW and the slew rate (SR). However, it will lead to higher power consumption and causes non-dominant poles to shift to lower frequencies, which may have a destabilizing effect. Moreover, the fixed bias current at the tail restricts the SR. These combined factors constrain the power efficiency of the RFC OTA circuit.

### 2.2. Concept of ERFC OTA

The proposed ERFC OTA structure is shown in Figure 2. $M_{3B}$, $M_{4B}$, $M_{9}$, and $M_{10}$ form a current mirror, which resembles the traditional current mirror OTA. This design includes several key modifications compared to the conventional structure depicted in Figure 1. It retains the cross-coupling structure to enhance dynamic characteristics and introduces a wide-swing, low-power adaptive tail current source for the input differential pairs $M_{1A}$, $M_{2A}$, $M_{1B}$, and $M_{2B}$. A nested local feedback circuit serves as the load for the first-stage amplifier. This network improves the transient response through its internal feedback loop. Furthermore, coupling capacitors $C_{BAT1}$ and $C_{BAT2}$, along with the bias resistor $R_{L}$, provide floating bias voltages for the gates of $M_{5}$–$M_{8}$. This addresses the potential operating point shift in traditional ERFC OTAs during the transient response.

MOSFETs biased in the weak inversion region achieve higher transconductance, reduced noise, and improved current efficiency at the cost of larger mismatch or area. The MOSFET current in the weak inversion region can be expressed as follows:(4)$$I_{D}=\left(\frac{W}{L}\right)I_{0}\exp\left(\frac{V_{GS}-V_{TH}}{mV_{T}}\right)\left[1-\exp\left(-\frac{V_{DS}}{V_{T}}\right)\right]$$
where W/L is the aspect ratio, $I_{0}=\mu C_{ox}V_{T}^{2}$ is a process-dependent constant, μ is the carrier mobility, $C_{ox}$ is the capacitance per unit area of the gate oxide, $V_{T}=kT/q$ is the thermal voltage, $V_{GS}$ is the gate-source voltage, *T* is the absolute temperature, *k* is Boltzmann’s constant, *q* is the electronic charge, $V_{TH}$ is the threshold voltage, *m* is the subthreshold slope factor, and $V_{DS}$ is the transistor drain-source voltage. When $V_{DS}$ is greater than 100 mV, Equation (Equation 4) can be simplified as follows:(5)$$I_{D}\approx \left(\frac{W}{L}\right)I_{0}\exp\left(\frac{V_{GS}-V_{TH}}{mV_{T}}\right)$$

Therefore, the transconductance of the transistor under subthreshold bias is expressed as follows:(6)$$g_{m}=\frac{\partial I_{D}}{\partial V_{GS}}=\frac{I_{D}}{mV_{T}}$$

As shown in Equations (Equation 5) and (Equation 6), MOSFETs operating under weak inversion exhibit a larger transconductance efficiency ($\frac{g_{m}}{I_{d}}$) compared to those operating in the saturation region. In this design, the circuit operates in the subthreshold region to enhance the OTA’s transient response while maintaining low power consumption. Circuit matching must be closely monitored to prevent mismatches from affecting the OTA’s static operating point and dynamic response.

## 3. Circuit Implementation

### 3.1. The ABNLF Technique

The proposed ABNLF technique consists of three parts: an adaptive wide-swing current source, a nested local feedback network, and an output stage floating bias circuit. The goal is to achieve a good transient response while maintaining low power consumption. Figure 3a illustrates the principle of adaptive biasing with input expansion. It uses two floating voltage sources, $V_{B}$, to bias the input differential pair. Figure 3b shows the transistor-level implementation of the floating voltage source bias circuit. $M_{1A}$, $M_{2A}$, $M_{1B}$, and $M_{2B}$ serve as the differential inputs for the designed OTA circuit. $M_{11}$ and $M_{12}$ act as the floating voltage sources, $V_{B}$, to provide a bias voltage to the differential pair. In the ideal case of a perfectly matched differential pair, the source stages of the differential pairs $M_{1A}$, $M_{2A}$, $M_{1B}$, and $M_{2B}$ match the source stages of the floating voltage sources $M_{11}$ and $M_{12}$, and the current of the input differential pair is 0.5$I_{B}$. Moreover, $M_{15}$ and $M_{16}$ are independently biased by the current source, $I_{C}$. Together with $M_{11}$, $M_{13}$, $M_{12}$, and $M_{14}$, they form a flipped voltage follower (FVF) structure. $M_{15}$ and $M_{16}$ are used as a level-shifter to increase the FVF input swing. Thus, when a large step signal is input to the differential pair, the adaptive biasing current source can provide a large dynamic current that is not limited by $I_{B}$. This current is positively correlated with the square of the input differential signal. Additionally, the transconductance of the input pair is effectively doubled by the cross-coupling of the two flipped voltage followers (FVFs), ensuring that the input transistors receive the fully differential signal.

The proposed nested-local feedback (NLF) is shown in Figure 4a. It is achieved by embedding the local positive feedback formed by the cross-coupling of $M_{17A}$ and $M_{18A}$ in the conventional LCMFB [15], where the transistor aspect ratio of $M_{3B}$:$M_{3A}$:$M_{17A}$ is 15:3:2. Additionally, the introduction of $M_{3C,4C,17B,18B}$ improves the replication accuracy of the current in static conditions. In the quiescent state, assuming the devices perfectly match, no current flows through the resistors $R_{1,2}=R_{A}$, and the voltage $V_{X}=V_{Y}=V_{Z}$. At this time, the static behavior of the NLF is similar to that described in [18].

In small-signal operation, when a fully differential current $i_{ind}$ is applied to the NLF, half of the current $i_{ind}/2$ flows through the resistors $R_{1,2}=R_{A}$, generating complementary voltages $v_{X}=-v_{Y}$ at nodes *X* and *Y*. Consequently, node *Z* can be treated as a virtual ground. The simplified small-signal model of the NLF is depicted in Figure 4b, where $g_{mi}$ and $r_{oi}$ represent the transconductance and output impedance of $M_{i}$, respectively. Thus, the impedance of nodes *X* and *Y* can be formulated as follows:(7)$$R_{X,Y}=\frac{R_{A}//r_{oeq,X,Y}}{1-g_{m17A,18A}\left(R_{A}//r_{oeq,X,Y}\right)}$$

It is important to feasibly design $g_{m17A,18A}$ and $R_{A}$ to ensure that $R_{X,Y}>0$, which guarantees the stability of the NLF. Additionally, the negative resistance $-1/g_{m17A,18A}$ introduced via the local positive feedback increases the resistance $R_{X,Y}$ at nodes *X* and *Y*, reducing the passive resistance area. The NLF current gain can be expressed as $A_{i}=g_{m3B,4B}R_{X,Y}$, which is independent of the current mirror ratio and is not affected by the parasitic capacitance at node *Z*.

Considering large-signal operation, if a large-signal current $I_{ind}$ is applied to the NLF, the voltages at nodes *X* and *Y* may become unbalanced due to the voltage drop across the LCMFB resistor $R_{A}$, and this instability is significantly reinforced by the local positive feedback loop. As a result, the voltage at node *X* will be very high while the voltage at node *Y* is close to 0. As a result, $M_{3B}$ enters saturation and the strong inversion region, while $M_{4B}$ and $M_{17A}$ enter the cutoff region, and $M_{18A}$ enters the deep linear region. Thus, the differential output current of the NLF is $I_{out}$ = $I_{o+}$−$I_{o-}$, and it can be expressed by the square law.

Taking an NMOS cascode transistor as an example, a conventional constant gate bias cascode transistor is shown in Figure 5a. When a large-signal current flows through the transistor, the voltage at its source decreases significantly, potentially shifting the circuit’s DC operating point. For example, in Figure 4a, $M_{5}$ is the cascode transistor with $M_{3B}$ as its source-connected current source, and $M_{3B}$ may enter the linear region under large-signal conditions, limiting the output current. The proposed floating-gate bias cascode transistor depicted in Figure 5b is implemented using a $R_{L}$ and $C_{BAT}$. In a static state, no current flows through the series $R_{L}$$C_{BAT}$, and $V_{C}$ is equal to $V_{E}$ without additional power consumption. If a large step is applied, an induced voltage is generated via an induction circuit and connected to $V_{S}$. $V_{S}$ travels through the RC highpass filter to the gate of the cascode transistor, preventing the transistor connected to its source from entering the linear region and increasing the permissible high current output. Furthermore, the induced voltage can be generated via the NLF without the need for additional circuits. The $R_{L}$$C_{BAT}$ is often designed as larger to achieve a lower low cut-off frequency. The $C_{BAT}$ does not need to be excessively large, as the local positive feedback loop significantly enhances the large-signal response at $V_{X,Y}$. This is primarily due to the amplified signal amplitude at $V_{S}$ introduced via the feedback mechanism, which reduces the dependency on a large $C_{BAT}$ to achieve the desired signal swing and response speed. Consequently, this design approach enables a more area-efficient implementation while maintaining robust performance under large-signal conditions. Moreover, the $R_{L}$ also can be realized using a transistor. This facilitates a reduction in the area.

### 3.2. Transistor-Level Implementation

Based on the proposed ABNLF technique, the final transistor-level circuit implementation of the ERFC OTA is depicted in Figure 6. In the proposed circuit, transistors $M_{15}$ and $M_{16}$ are biased in the saturation region to achieve a higher source-to-gate voltage ($V_{SG}$), which facilitates an extended input range under the given bias current, $I_{C}$. Transistors $M_{19}$ to $M_{21}$ are operated in the deep triode region, functioning as active resistors to provide the desired impedance characteristics. All remaining transistors are biased in the subthreshold region to maximize the intrinsic gain and transconductance ($g_{m}$) for a given bias current, thereby enhancing the overall performance of the circuit. This strategic biasing approach ensures optimal trade-offs between input range, gain, and power efficiency.

Table 1 shows the design dimensions of the proposed ERFC OTA and traditional RFC OTA. In the design and optimization of the ERFC OTA, the selection of transistor dimensions is a critical design decision, directly impacting the circuit’s performance, power consumption, and stability. For the differential pair and cascode transistors, a large width-to-length ratio and a short channel length are employed to enhance the gain bandwidth (GBW) and input transconductance. For the current mirror, a longer channel length is preferred to increase the output impedance. Additionally, the area of the differential pair is carefully balanced to further optimize input noise performance. In the ERFC OTA design, the transistor dimensions must be carefully traded off among gain, power consumption, noise, stability, robustness, and area. By optimizing the width and length of the transistors, a high-performance, low-power, and robust OTA can be achieved, meeting the demands of complex analog circuit design.

Compared with the traditional RFC OTA, the proposed ERFC OTA needs to consider the proper operation of the FVF (folded-voltage-follower) structure, which sacrifices some voltage margin. The proposed ERFC OTA utilizes the FVF structure with input extension to replace the adaptive biasing of the input-pair transistor $M_{0}$. For proper operation, the FVF transistors ($M_{11}/M_{12}$, $M_{13}/M_{14}$) must remain in the saturation region to maintain the appropriate operating point. Under small input conditions, transistors $M_{11}$ and $M_{12}$ enter the linear region, defining the lower limit of the input signal. Conversely, under large input conditions, transistors $M_{13}$ and $M_{14}$ enter the linear region, causing a significant decrease in the gate-to-source voltages of $M_{15}$ and $M_{16}$. This drives the current source, $I_{B}$, into the linear region, disrupting the operating point and defining the upper limit of the input signal. Therefore, the input voltage range of the proposed traditional ERFC OTA is expressed as follows:(8)$$V_{DD}-V_{SG11}-V_{SG13}-V_{SD11,sat}-V_{SG15}\leq V_{in}\leq V_{DD}-V_{SG11}-V_{SD13,sat}$$

Since the output branches of the traditional RFC OTA and the proposed ERFC OTA are almost the same, and the floating-gate bias does not affect its output signal range, the output signal ranges are identical and can be expressed as follows:(9)$$V_{DS4_{B,sat}}+V_{DS6,sat}\leq V_{out}\leq V_{DD}-V_{SD10,sat}-V_{SD8,sat}$$

### 3.3. Small-Signal Analysis

The small-signal half-circuit of the ERFC OTA is illustrated in Figure 7. The transconductance and GBW of the proposed ERFC OTA are expressed as follows:(10)$$g_{m,\mathrm{ERFC}}=2\left(g_{m1B}g_{m4B}R_{Y}+g_{m2A}\right)$$(11)$${GBW}_{\mathrm{ERFC}}=\frac{2(g_{m1B}g_{m4B}R_{Y}+g_{m2A})}{2\pi C_{L}}$$
where $R_{Y}\approx R_{1,2}/\left(1-g_{m17A,18A}R_{1,2}\right)$ and $R_{C}\approx 1/g_{m5,6}$, respectively.

Factor 2 in Equation (Equation 11) is attributed to the transconductance provided via the adaptive bias circuits of the differential pair, and factor $g_{m4B}R_{Y}$ is attributed to the NLF configuration.

The RFC and ERFC OTA have identical output branches in terms of quiescent current and transistor aspect ratios. Consequently, the output impedance is also the same and is expressed as follows:(12)$$Rout\approx \left(r_{o2A}//r_{o4B}\right)g_{m6}r_{o6}//r_{o10}g_{m8}r_{o8}$$

Stability is a fundamental requirement for the proper operation of an OTA. It is primarily governed by the phase margin of the amplifier, which is directly influenced by the positions of the poles and zeros in the transfer function. For the same load capacitance, $C_{L}$, since the output impedances of both the ERFC OTA and the conventional RFC OTA are identical and very large, they share the same dominant pole, which can be expressed as $w_{p1}=-1/R_{out}C_{out}\approx -1/R_{out}C_{L}$. The non-dominant poles of the ERFC OTA are $w_{p2}=-1/R_{X,Y}C_{x,Y}$ and $w_{p3}=-1/R_{C}C_{C}$, where $C_{X,Y}\approx C_{gs3B,4B}+C_{gs18A,17A}$ and $C_{C}$ represents the parasitic capacitance at the source node of $M_{5}$ or $M_{6}$. Given that $R_{X,Y}\cdot C_{X,Y}\gg R_{C}\cdot C_{C}$, the first non-dominant pole $w_{p2}$ of the proposed ERFC OTA must be at least 2.2 times larger than the unity-gain–bandwidth (GBW) to ensure a sufficient phase margin. Typically, the amplifier phase margin can be expressed as follows: $PM=180-\tan^{-1}\left(GBW/w_{p1}\right)-\tan^{-1}\left(GBW/w_{p2}\right)-\tan^{-1}\left(GBW/w_{p3}\right)$. Since $w_{p1}\ll GBW$, it follows that $\tan^{-1}\left(GBW/w_{p1}\right)\approx 90^{\circ }$. Additionally, since $w_{p3}\gg w_{p2}$, and $\tan^{-1}\left(1/2.2\right)\approx \tan^{-1}\left(0.4545\right)\approx 24.4^{\circ }$. Therefore, the PM of the proposed ERFC OTA is given as follows:(13)$$\begin{array}{cc} \; & PM\approx 90^{\circ }-\tan^{-1}\left(\frac{GBW}{f_{X,Y}}\right)\\ \; & \approx 90^{\circ }-\tan^{-1}\left\{\frac{2\left[g_{m,1A}g_{m,3B}R_{X,Y}+g_{m,1A}\right]R_{X,Y}C_{X,Y}}{C_{L}}\right\} \end{array}$$

Therefore, the required load capacitance can be described as follows:(14)$$C_{L,\min}\geq \frac{2\left(g_{m,1A}g_{m,3B}R_{X,Y}+g_{m,1A}\right)R_{X,Y}C_{X,Y}}{0.45}$$

In this design, a 70 pF capacitor will be used as the compensation capacitor $C_{L}$, sacrificing the gain in bandwidth to ensure loop stability.

### 3.4. Noise Analysis

Flicker noise and thermal noise are the primary noise sources in CMOS analog circuits. The input-referred mean-square noise voltage spectral density in a CMOS device is defined as follows:(15)$$\bar{V_{n}^{2}}\left(f\right)=\frac{4kT\gamma }{g_{m}}+\frac{K_{P/N}}{C_{ox}WLf}$$
where *k* is the Boltzmann’s constant, *T* is the absolute temperature, and the factor γ varies from 1/2 to 2/3, e.g., from weak to strong inversion, for long-channel devices. $K_{P}{/}_{N}$ represent the flicker noise coefficient, which is dependent on the CMOS process and device characteristics. The first and second terms of Equation (Equation 15) represent the thermal and flicker noise spectral density, respectively. Similarly, the thermal noise spectral density for a resistor of resistance R is given as follows:(16)$$\bar{V_{nR}^{2}}=4kTR$$

Under the assumption that all noise sources are uncorrelated and, for simplicity, that we have the same γ factor for all MOSFETs, the input-referred mean-square thermal noise spectral density of the RFC OTA in a bandwidth Δf can be expressed as follows:(17)$$\bar{V_{T,inRFC}^{2}}=\frac{8kT\gamma \left[\left(1+K^{2}\right)+\left(1+K\right)\frac{g_{m3A}}{g_{m1A}}+\frac{g_{m9}}{g_{m1A}}\right]}{g_{m1A}{\left(1+K\right)}^{2}}$$

The thermal noise expression for the proposed ERFC OTA can be expressed as Equation (Equation 18).(18)$$\bar{V_{T,inERFC}^{2}}=\frac{2kT\gamma }{g_{m1A}{\left(1+g_{m3B}R_{X,Y}\right)}^{2}}\left[\left(1+g_{m3B}^{2}R_{X,Y}^{2}\right)+g_{m3B}^{2}R_{X,Y}^{2}\left(1+\alpha +\frac{1}{\gamma g_{m3A}R_{1,2}}\right)\frac{g_{m3A}}{g_{m1A}}+\beta \frac{g_{m3A}}{g_{m1A}}+\frac{g_{m9}}{g_{m1A}}\right]$$
where α and β are the ratios of $M_{17A}$ and $M_{3A}$, as well as $M_{3B}$ and $M_{3A}$, which are 2/3 and 15/3, respectively.

For the flicker noise, it is again assumed that all noise sources are uncorrelated and equal for the same type (NMOS or PMOS) of transistor $K_{P}{/}_{N}$. The input-referred mean-square flicker noise of the RFC OTA in a bandwidth Δf is shown as Equation (Equation 19). The expression of flicker noise for the proposed ERFC OTA is Equation (Equation 20).

Determining which configuration has lower noise directly based on Equations (Equation 17)–(Equation 20) is difficult. However, by optimizing the transconductance and size of the input MOSFETs for the proposed ERFC OTA, we can minimize the input-referred noise. This optimization enables the ERFC OTA to achieve lower noise than traditional designs, primarily due to its larger equivalent input transconductance, which reduces input-referred noise.(19)$$\bar{V_{F,in}^{2}}\left(f\right)=\frac{2\left[\frac{K_{P}}{C_{ox}{\left(WL\right)}_{1A}f}g_{m1A}^{2}+\frac{K_{P}}{C_{ox}{\left(WL\right)}_{1A}f}g_{m1A}^{2}K^{2}+\frac{K_{N}}{C_{ox}{\left(WL\right)}_{3A}f}g_{m3A}^{2}K^{2}+\frac{K_{N}}{C_{ox}{\left(WL\right)}_{3B}f}g_{m3B}^{2}+\frac{K_{P}}{C_{ox}{\left(WL\right)}_{9}f}g_{m9}^{2}\right]}{{\left(1+K\right)}^{2}g_{m1A}^{2}}$$(20)$$\bar{V_{F,inERFC}^{2}}\left(f\right)=\frac{2\left\{\frac{K_{P}}{C_{ox}{\left(WL\right)}_{1A}f}g_{m1A}^{2}\left[1+g_{m3B}^{2}R_{X,Y}^{2}\right]+g_{m3B}^{2}R_{X,Y}^{2}\left[\frac{K_{N}}{C_{ox}{\left(WL\right)}_{3A}f}g_{m3A}^{2}+\frac{K_{N}}{C_{ox}{\left(WL\right)}_{17A}f}g_{m17A}^{2}\right]+\frac{K_{N}}{C_{ox}{\left(WL\right)}_{3B}f}g_{m3B}^{2}+\frac{K_{P}}{C_{ox}{\left(WL\right)}_{9}f}g_{m9}^{2}\right\}}{4g_{m1A}^{2}{\left(1+g_{m3B}R_{X,Y}\right)}^{2}}$$

### 3.5. Large-Signal Response

The slew rate (SR) is an important factor in the settling time of the proposed ERFC OTA. Under static conditions, the currents of $M_{1A,1B}$ and $M_{2A,2B}$ are all equal to $0.5I_{B}$. The currents flowing into nodes *X* and *Y* are $I_{in+}$ and $I_{in-}$, provided via $I_{2B}$ and $I_{1B}$, respectively. They are also equal to $0.5I_{B}$ and distributed via transistors $M_{17A}:M_{3A}$ in a 2:3 ratio. Therefore, the current of $M_{3A}$ is $0.3I_{B}$, and no current flows through resistor $R_{1,2}$. The voltages of nodes of *X*, *Y*, and *Z* are $V_{X}=V_{Y}=V_{Z}=mV_{T}\ln\left\{{0.3I}_{B}/\left[{\left(W/L\right)}_{3A,4A}I_{o}\right]\right\}+V_{TH}$.

We assume that the large-signal $V_{id}=\left(V_{in+}-V_{in-}\right)>0$ is applied to the input pair. The operating region of $M_{2B}$ shifts from weak inversion at static to strong inversion, while the operating region of $M_{1B}$ shifts from weak inversion at static to the cut-off region. Therefore, $M_{2B},M_{1B}$ yields a differential current, $I_{ind}=I_{in+}-I_{in-}\approx I_{2B}$, which can be expressed as follows:(21)$$\begin{array}{cc} I_{2B}= & \frac{1}{2}\beta_{2B}{\left(mV_{T}\ln\left[\frac{0.5I_{B}}{I_{0}\left(W/L\;\right)}\right]+V_{id}\right)}^{2}\\ \; & \approx \frac{1}{2}\beta_{2B}V_{id}^{2} \end{array}$$

The NLF generates a significant voltage swing at nodes *X* and *Y*, resulting in a current of $I_{ind}/2$ through both resistors. This local positive feedback accelerates the voltage imbalance at these nodes. The current through $M_{3A}$ and $M_{4A}$ is the common-mode component, $I_{cm}\approx \left(I_{in+}+I_{in-}\right)/2-\left(I_{17A}+I_{18A}\right)/2\approx I_{2B}/2>>0.3I_{B}$. Hence, the voltage of *X*, *Y*, and *Z* can be given as follows:(22)$$\left\{\begin{array}{cc} \; & V_{Z}=\sqrt{\frac{2I_{cm}}{\beta_{3A,4A}}}+V_{TH}\\ \; & V_{X}=V_{Z}+\frac{R_{1}I_{ind}}{2}\\ \; & V_{Y}=V_{Z}-\frac{R_{2}I_{ind}}{2} \end{array}\right.$$

The ERFC output current $I_{out}$ is represented as shown below:(23)$$\begin{array}{cc} I_{out}\approx I_{o+} & =I_{3B}=\frac{1}{2}\beta_{3B}{\left(\sqrt{\frac{2Icm}{\beta_{3A}}}+\frac{R_{1}I_{ind}}{2}\right)}^{2}\\ \; & \approx \frac{1}{2}\beta_{3B}{\left(\sqrt{\frac{\beta_{2B}}{2\beta_{3A}}}V_{id}^{2}+\frac{R_{1}\beta_{2B}}{4}V_{id}^{2}\right)}^{2} \end{array}$$

From Equation (Equation 23), $I_{out}$ is proportional to $V_{id}^{4}$. A similar result occurs during negative slewing. Therefore, the slew rate (SR) of the proposed ERFC OTA can be expressed as follows:(24)$$SR_{ERFC}=\frac{I_{out}}{C_{L}}$$

By integrating the Class-AB input stage with the nonlinear feedback (NLF) mechanism, the proposed enhanced recycling folded cascode (ERFC) OTA achieves a maximum output current proportional to $V_{id}^{4}$, significantly enhancing the slew rate (SR) compared to the conventional recycling folded cascode (RFC) OTA.

## 4. Post-Layout Simulations and Comparisons

The proposed ERFC OTA uses an 0.18 μm CMOS process with a supply voltage of 1.8 V and a quiescent current of 10.4 μA. The bias current $I_{B}$ is 2 μA, and the load capacitance $C_{L}$ is 70 pF. The layout of the proposed ERFC OTA is presented in Figure 8, with an active area of 107 × 83 μ
$m^{2}$.

The open-loop frequency response of two OTAs is shown in Figure 9. The post-layout simulation uses a typical corner and a common-mode input voltage of 800 mV. The proposed ERFC OTA achieves a GBW of 1.37 MHz, six times higher than the RFC OTA. The DC gain of the ERFC OTA is 84.18 dB, which is significantly greater than that of the RFC OTA. The phase margin (PM) for the ERFC OTA is $67.05^{\circ }$, and the PM for the RFC OTA is $89.15^{\circ }$. The ERFC OTA’s PM decreases by $22.1^{\circ }$ due to increased transconductance. However, it remains well above the $60^{\circ }$ threshold, ensuring stable operation.

Figure 10 illustrates the noise performance of the two amplifiers. At 1 MHz, the input-referred noise, primarily dominated by thermal noise, is 31.86 nV/$\sqrt{\mathrm{Hz}}$ for the proposed ERFC OTA and 37.77 nV/$\sqrt{\mathrm{Hz}}$ for the RFC OTA. At 0.1 Hz, where flicker noise is the dominant factor, the input-referred noise is 26.61 μV/$\sqrt{\mathrm{Hz}}$ for the proposed ERFC OTA and 40.10 μV/$\sqrt{\mathrm{Hz}}$ for the RFC OTA. The proposed amplifier demonstrates low noise performance due to its enhanced transconductance.

Two OTAs are configured as unity-gain-inverting amplifiers to simulate low- and large-signal transient responses. A 100 kHz, a 10 mV peak-to-peak periodic square wave with a 50% duty cycle and an 800 mV DC level is applied. The output transience is illustrated in Figure 11. The ERFC OTA achieves a 1% settling time of 0.473 μs, which is significantly faster than the RFC OTA’s 2.945 μs. This improvement is attributed to increased transconductance.

The proposed ERFC OTA demonstrates significantly improved large-signal transient performance compared to the RFC OTA, as shown in Figure 12. When a 600 mV peak-to-peak periodic square wave is applied, the ERFC OTA achieves an average slew rate of 19.32 V/μs, which is better than the RFC OTA’s 0.16 V/μs. Additionally, the ERFC OTA’s step response shows no ringing, indicating a minimal impact on PM. Although there is an overshoot of approximately 100 mV, it is smaller than that reported in prior work [24].

Table 2 compares the performance parameters of the designed ERFC OTA with conventional RFC OTAs. Table 3 shows the simulation results across process, power supply voltage, and temperature (PVT) variations. The results indicate that the ERFC OTA has an excellent GBW and transient response while using nearly the same static power consumption.

To evaluate the effects of process variations and mismatches on the proposed ERFC OTA, 200-point Monte Carlo simulations were performed, and the simulation results are shown in Figure 13. The GBW, DC gain, phase margin, and offset voltage under 200-point Monte Carlo simulation are shown in Figure 13a,b,c and d, respectively. The mean values of GBW, DC gain, phase margin, and offset are 1.376 MHz, 84.15 dB, $66.77^{\circ }$, and 0.036 mV, respectively. The standard deviations are 0.18 MHz, 1.133 dB, $34.20^{\circ }$, and 1.094 mV, respectively. The proposed ERFC OTA maintains stable performance even in subthreshold regions.

The well-known figures of merit ($FoM_{S}$ and $FoM_{L}$) are used to quantitatively compare OTA performance and evaluate the power efficiency of low and large signals, respectively. They can be expressed as follows [23]:(25)$${FoM}_{S}=\frac{GBW\cdot C_{L}}{I_{supply}}$$(26)$${FoM}_{L}=\frac{SR\cdot C_{L}}{I_{supply}}$$

The comparison with prior works is shown in Table 4. The proposed ERFC OTA achieves a GBW of 1.37 MHz, surpassing the 1.12 MHz reported in [24] while consuming less power. While its GBW is lower than [7,8], it consumes significantly less supply current. The proposed OTA also demonstrates a slew rate of 19.32 V/μs at a current consumption of 10.4 μA, outperforming [11,24]. Furthermore, the proposed ERFC OTA achieves $FoM_{S}$ and $FoM_{L}$ values of 9.22 and 130.04, respectively, which are substantially higher than those of other works listed in Table 4, highlighting its superior power efficiency. It is particularly suitable for applications that require less power while achieving a high performance, such as portable devices and IoT systems.

## 5. Conclusions

This work has presented innovative enhancements to the traditional RFC OTA, introducing the ABNLF technique, which effectively addresses the power consumption-versus-transient response trade-off in amplifiers through nested local feedback and adaptive biasing. Compared to conventional RFC OTA, it incurs only a 30% increase in power consumption while boosting the SR and GBW by 120- and 5.95-fold, respectively. It is poised to be an effective solution in low-power amplifier applications.

## Figures and Tables

**Figure 1 sensors-25-02523-f001:**
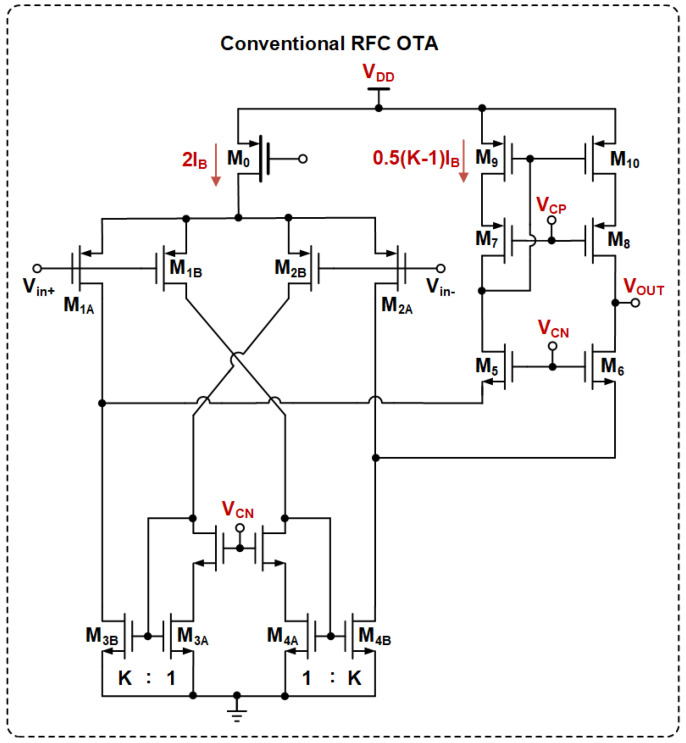
Conventional RFC OTA [13,14].

**Figure 2 sensors-25-02523-f002:**
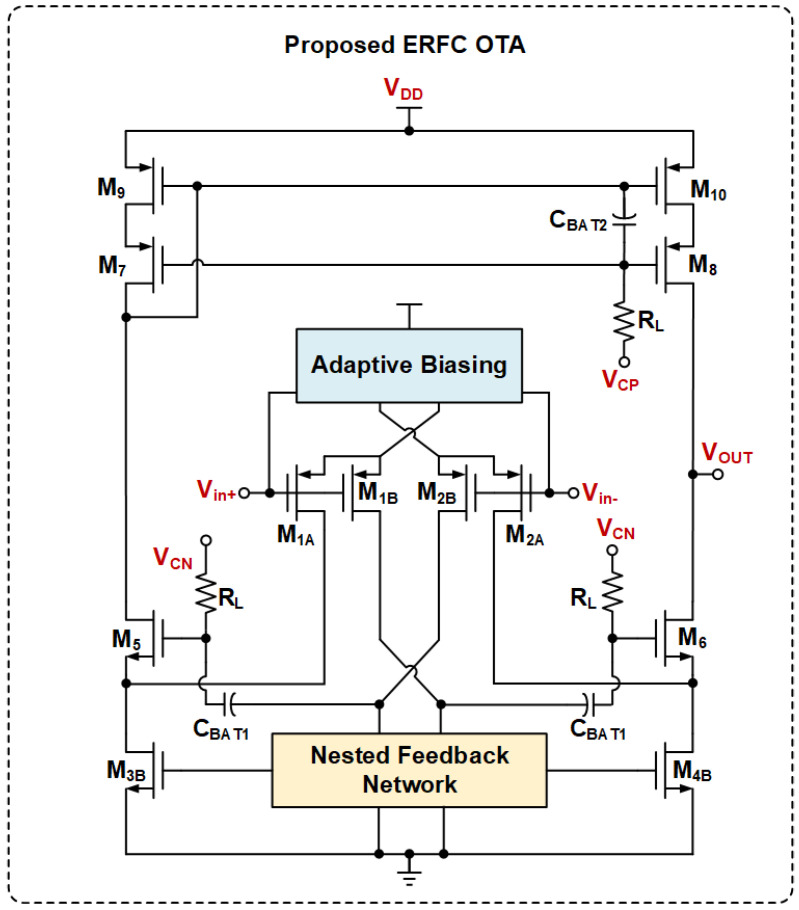
Concept of the proposed ERFC OTA.

**Figure 3 sensors-25-02523-f003:**
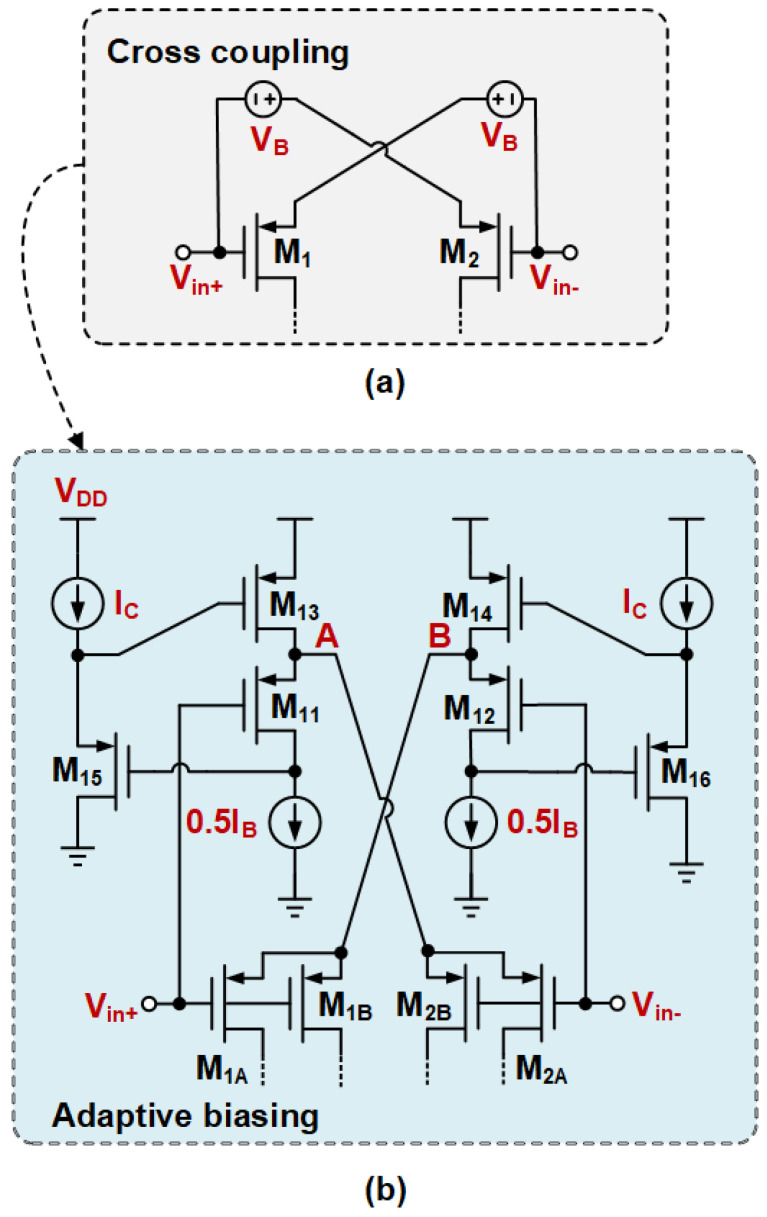
Adaptive bias with input expansion.

**Figure 4 sensors-25-02523-f004:**
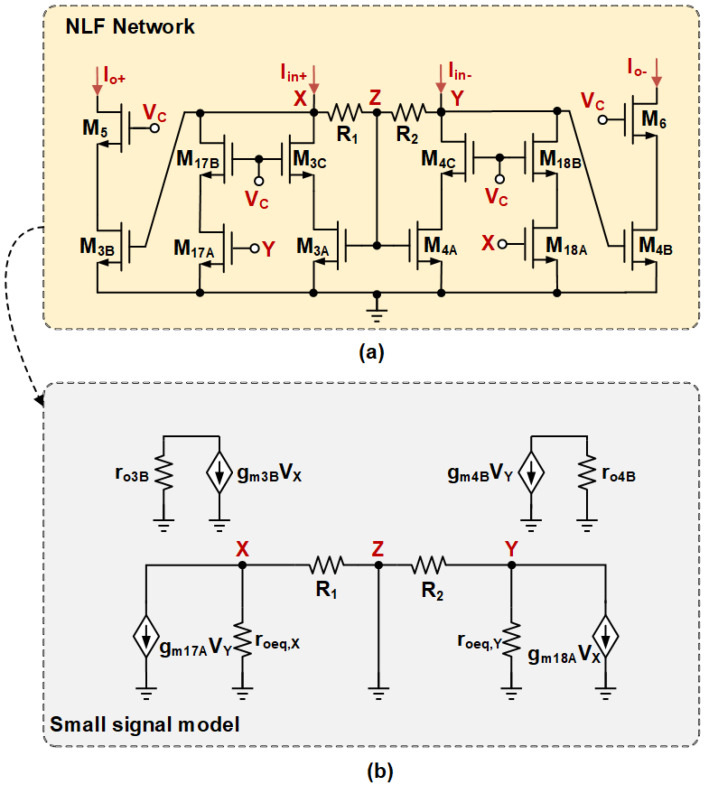
Proposed nested local feedback (**a**) circuit with (**b**) as mall-signal model.

**Figure 5 sensors-25-02523-f005:**
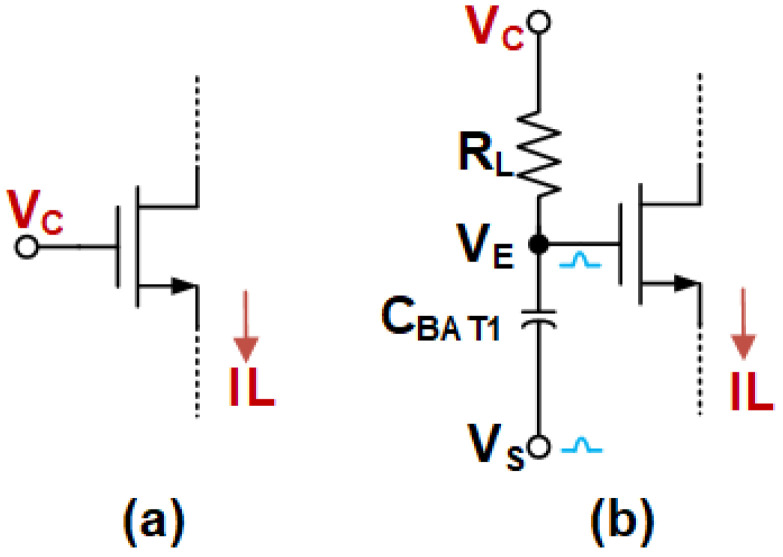
The bias cascode transistor in (**a**) constant and (**b**) floating.

**Figure 6 sensors-25-02523-f006:**
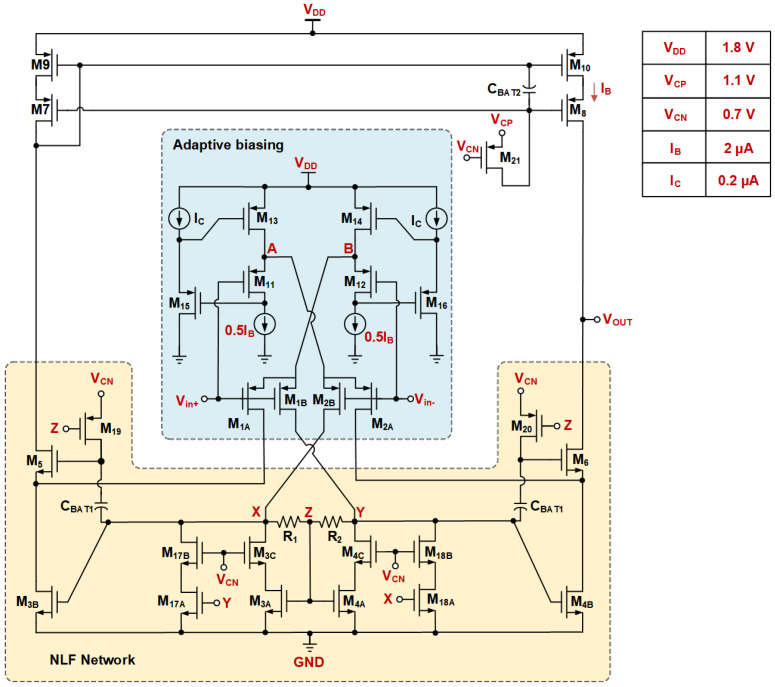
Transistor-level implementation of proposed ERFC OTA.

**Figure 7 sensors-25-02523-f007:**
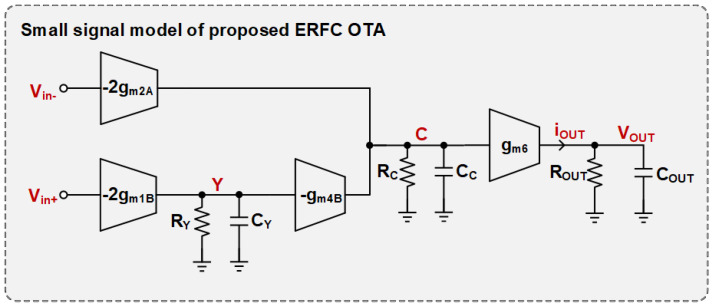
The small-signal half-circuit of ERFC OTA.

**Figure 8 sensors-25-02523-f008:**
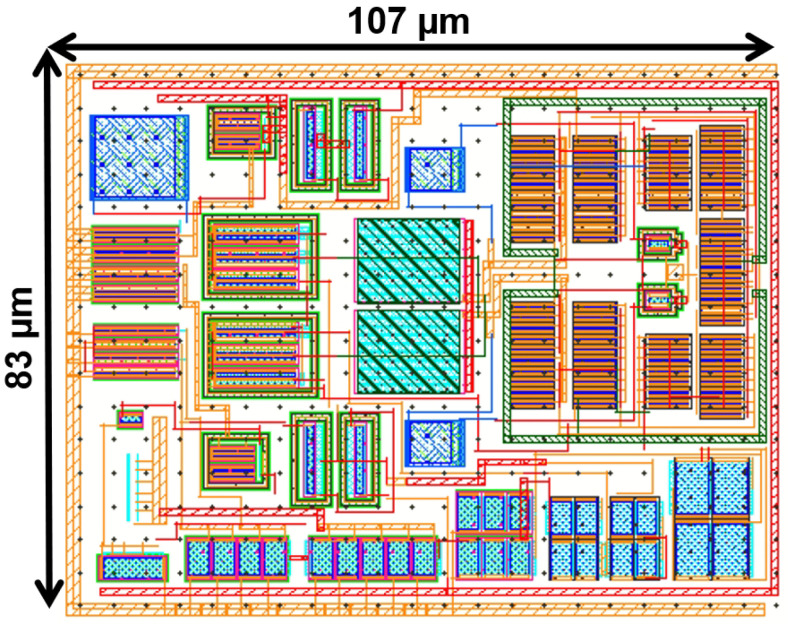
The layout of ERFC OTA.

**Figure 9 sensors-25-02523-f009:**
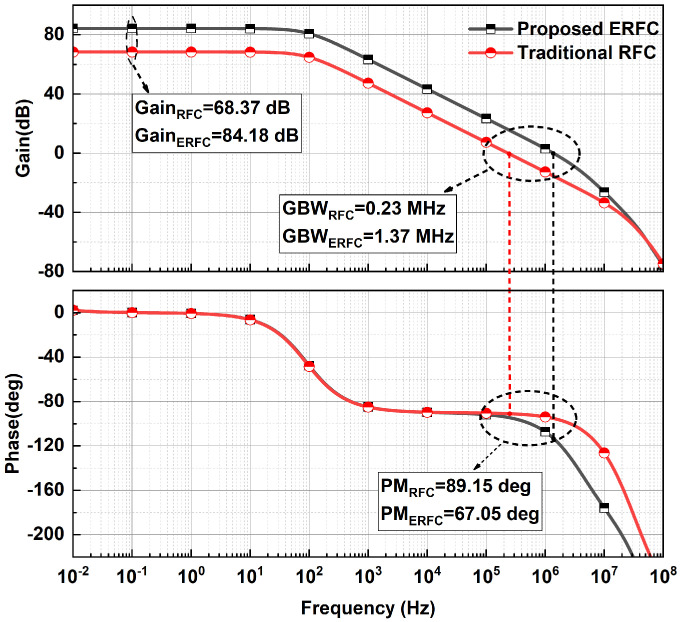
Open-loop frequency response of ERFC OTA.

**Figure 10 sensors-25-02523-f010:**
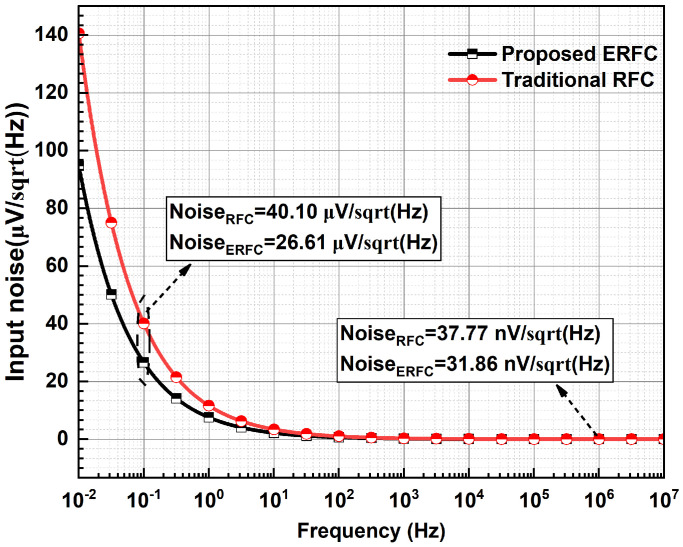
Equivalent input noise of ERFC OTA.

**Figure 11 sensors-25-02523-f011:**
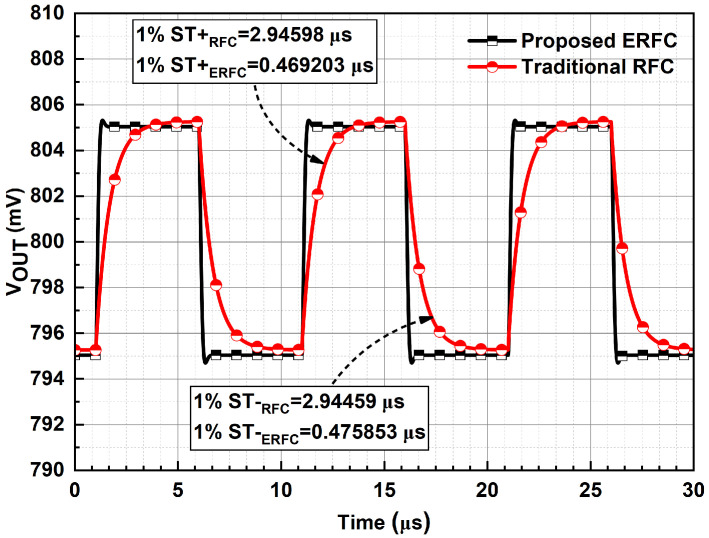
Small-signal transient response of ERFC OTA.

**Figure 12 sensors-25-02523-f012:**
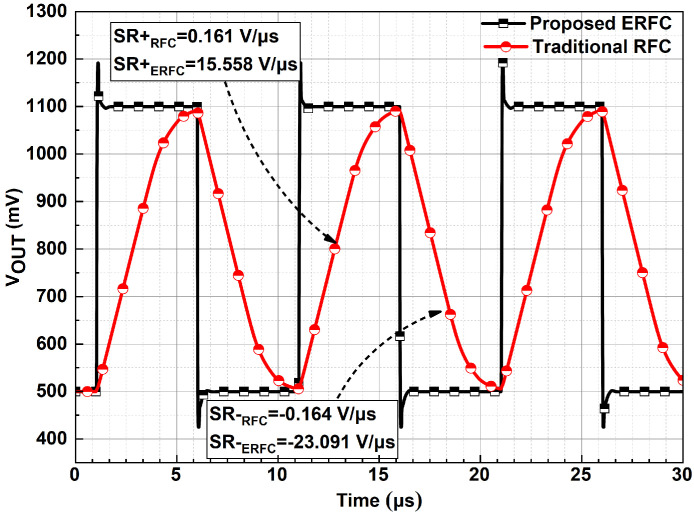
Large-signal transient response of ERFC OTA.

**Figure 13 sensors-25-02523-f013:**
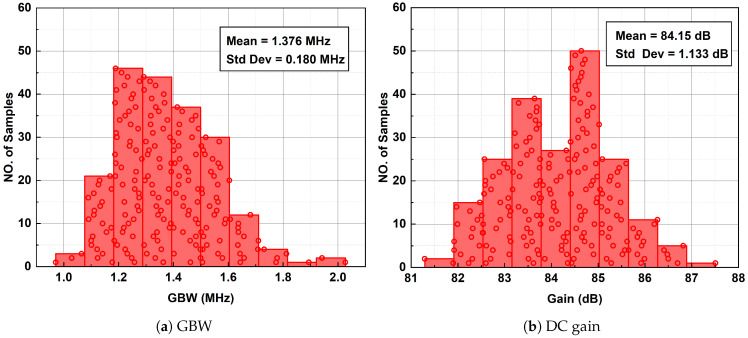
Two hundred-point Monte Carlo simulation.

**Table 1 sensors-25-02523-t001:** OTA device dimensions under PVT variations.

ERFC OTA				RFC OTA	
$M_{1A,1B,2A,2B,11,12}$	24 μm/0.6 μm	$M_{9,10}$	48 μm/0.3 μm	$M_{1A,1B,2A,2B}$	24 μm/0.6 μm
$M_{3A,4A}$	18 μm/0.3 μm	$M_{13,14}$	24 μm/0.3 μm	$M_{3A,4A}$	18 μm/0.3 μm
$M_{17A,18A}$	12 μm/0.3 μm	$M_{3C,4C,17B,18B}$	30 μm/0.3 μm	$M_{3B,4B}$	90 μm/0.3 μm
$M_{3B,4B}$	90 μm/0.3 μm	$M_{15,16}$	1 μm/20 μm	$M_{7,8}$	24 μm/0.3 μm
$M_{5,6}$	60 μm/0.3 μm	$M_{19,20,21}$	3 μm/1 μm	$M_{5,6}$	60 μm/0.3 μm
$M_{7,8}$	24 μm/0.3 μm	$R_{1,2}$	52 kΩ	$M_{9,10}$	48 μm/0.3 μm

**Table 2 sensors-25-02523-t002:** The performance comparison of two OTAs.

Parameter	Traditional RFC OTA	Proposed ERFC OTA
Supply current (μA)	8	10.4
$C_{L}$ (pF)	70	70
GBW (MHz)	0.23	1.37
DC gain (dB)	68.37	84.18
Phase margin (^∘^)	89.15	67.05
Average slew rate (V/μs)	0.16	19.32
1% setting time (μs)	2.945	0.473
Input noise (nV/$\sqrt{\mathrm{Hz}}$)	37.77	31.86
$FoM_{S}$ (MHz pF/μA)	2.01	9.22
$FoM_{L}$ (V/μs) pF/μA)	1.40	130.04

**Table 3 sensors-25-02523-t003:** Performance of the OTA under PVT variations.

Parameter	Corner					V_DD_		Temp	
	**FF**	**SS**	**SF**	**FS**	**TT**	**−10%**	**+10%**	**−40°**	**+125°**
GBW (MHz)	1.71	1.10	1.41	1.33	1.37	1.14	1.63	1.22	1.47
PM (^∘^)	63.29	69.03	65.95	67.45	66.86	72.32	60.95	70.42	57.87
Gain (dB)	83.87	84.02	84.87	83.55	84.18	82.56	85.7	84.45	82.48
SR_*av*_ (V/μs)	18.74	18.015	17.81	19.79	19.175	18.42	19.435	15.68	18.005
ST_*av*_ (ns)	391.3	539.3	468.25	479.5	471.95	359.65	429.2	388.15	626.85

**Table 4 sensors-25-02523-t004:** Comparison with prior arts.

Parameter	[7]	[8]	[11]	[12]	[24]	This Work
Supply voltage (V)	1.8	1.8	1	2	0.8	1.8
Technology (nm)	180	180	180	500	180	180
Supply current (μA)	344	800	50	50	45	10.4
Load (pF)	10	5.6	20	70	130	70
DC gain (dB)	72.7	60.9	92.3	72.7	102.7	84.18
GBW (MHz)	173.3	134.2	6.51	3.4	1.12	1.37
Phase margin (^∘^)	55	70.6	75.81	75.1	67.85	67.05
Slew rate (V/μs)	139.4	94.1	15.19	19.25	1.033	19.32
1% Settling time (ns)	9.2	10.2	79.5	110	555	473
Offset voltage (mV)	7.3	7.6	N/A	N/A	3.72	3.3
Noise 1 MHz (nV/$\sqrt{\mathrm{Hz}}$)	48	N/A	N/A	23	68.8	31.9
$FoM_{S}$ (MHz·pF/μA)	5.03	0.94	2.604	4.76	3.326	9.22
$FoM_{L}$ ((V/μs)·pF/μA)	4.07	0.66	6.07	26.95	2.98	130.04
Measured/simulated	Meas.	Sim.	Sim.	Meas.	Meas.	Sim.

## Data Availability

Data will be made available upon request.

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
