# Peer review of "Power-Efficient Recycling Folded Cascode Operational Transconductance Amplifier Based on Nested Local Feedback and Adaptive Biasing"

_sensors, 2025, doi:10.3390/s25082523_

Round 1

Reviewer 1 Report

Comments and Suggestions for Authors This paper shows how to improve transient response by increasing the transconductance of the input and slew late by using proposed adaptive bias circuitry and nested local feedback techniques in the design of the OP amplifier. This technique improves the slew rate by a factor of 120 and the gain-bandwidth by a factor of 6. In the paper, the background of the research, the means of improving the circuit, the circuit operation, and the prediction performance estimation are theoretically sufficiently described, and the effect is sufficiently verified by simulation with LPE after layout. PVT fluctuations are also shown by corner analysis, indicating that the proposed circuit is practically viable. Therefore, it is judged to be worthy of publication as an academic paper. However, since equations (16), (17), and (18) are missing, it is necessary to add this part to make it a complete paper. It should be published after this amendment. The authors should add the equations (16), (17), and (18).

Reviewer 2 Report

Comments and Suggestions for Authors

The paper was written carelessly, and the English should be thoroughly revised. Major corrections are suggested, and the paper should be returned for another round of revisions. Among other things, the following points need to be addressed:

  • Define ERFC the first time it is mentioned.
  • Line 119: Grammatical error ("matche").
  • Error in the legend of Table 3.
  • Equations are not numbered sequentially.
  • Line 217: Vague statement— "This optimization (?) allows the ERFC OTA to achieve lower noise levels compared to traditional designs." Clarify the meaning.
  • Line 240: Incomplete statement.
  • Lines 226–229: The sentence is unclear and does not make sense. Rewrite for clarity.
  • References are not cited in order.
  • Include a section titled "Design Considerations" discussing the rationale behind the transistor sizing.
  • Provide a table with the transistor sizes for both the RFC and ERFC.
  • Figure 6: Include information about the power supplies and operating points in the schematic.
  • Figure 6: Are all transistors operating in the weak inversion region? If not, specify which ones are and which ones are not.
  • Figure 6: How can you ensure that the transistors operate in weak inversion for any input/output signal amplitude? Provide a detailed discussion.
  • Compare the input linear range of the RFC and the ERFC.
  • Compare the output linear range of the RFC and the ERFC.
  • Equation (11): Verify the correctness of PM. Typically, PM = 180 - atan( ).
  • Table 1: The phase margin appears good for both the RFC and ERFC due to the large load capacitance (CL = 70 pF). However, the load capacitance should represent the input capacitance of the next stage. Values around 1 pF should be sufficient. Re-simulate and compare the RFC and ERFC with CL = 1 pF.
  • Figure 8: Indicate the location of CL in the layout. Pay special attention to the PM.
  • Figure 13: Add the same plots for the RFC to enable a direct comparison between the RFC and ERFC within the same figure.
  • The statement "boosting the rise time and GBW by 120 times" is not supported by the provided data. A detailed discussion justifying this claim must be included.

Additionally:

  • In the introduction, it is unclear why the design is focused on low-power applications. If your proposed topology (Figure 6) is designed for high-frequency applications, do your claims remain valid?
  • The statement: "The results indicate that the ERFC OTA has excellent gain-bandwidth product (GBW) and transient response while using nearly the same power consumption." What type of power consumption are you referring to? Does your claim hold for dynamic power consumption?
Comments on the Quality of English Language

The entire document must be reviewed and rewritten.

Reviewer 3 Report

Comments and Suggestions for Authors

This work presents a modified recycling cascode OTA that uses a combination of adaptive biasing, nested local feedback and floating bias techniques to enhance the circuit GBW and significantly increase its slew rate performance. The article is well organized and the conclusions are backed up by post-layout simulation results. Therefore, i recommend the article's acceptance with minor revisions (mostly phrasing improvements and typos, see below).

Comments on the content:

  1. L92: Usually mosfets operating in weak inversion have a large W/L ratio compared to strong inversion. Therefore the statement "at the cost of larger mismatch"  may not be always valid.
  2. L103: Mosfets operating in weak inversion exhibit larger transconductance efficiency (gm/id).
  3. L123: This design aims to increase the input swing of Vsg15,16. This sentence is wrong. M15,16 are used as a level-shifter to invrease the FVF input swing by approximately Vsg15,16.
  4. L131: I believe that reference [14] is wrong here. Please cross-check and provide the correct reference.
  5. L152: As a result... M3A should be replaced by M3B. M3A at the end of the line should be replaced by M4B.
  6. L168: "The Cbat...of the Vx,y". Please elaborate more on this statement so that the meaning is clear (e.g. is it due to the large amplitude at Vs due to the local feedback that Cbat can be small?).
  7. L189: Cc=Cgs5,6. This would require the gate of M5,6 to be low impedance (virtual ground), but this is not exactly true.
  8. L191: wp2>2.2GBW is true for specifically 60 deg PM. Also, Please specify in the text the target PM that corresponds to eq. 12. According to my calculations eq. 12 corresponds to 30 deg PM, which is low.
  9. L217: "This optimization...traditional designs". How is this statement justified?
  10. L271: Conventional OTAs -> conventional RFC OTA.

Comments on the Quality of English Language
  1. L72: With -> When
  2. L73: "by the bias currents 2IB and K". K is not a bias current. Please rephrase
  3. L119: matche -> match
  4. L154: ...as the square law -> by the square law
  5. L191: OTA must be at least....
  6. L222: which are provided...
  7. L222: "...respectively. It is also...". Incorrect phrasing. Please rewrite the sentence.
  8. L224: "The nodes...+VTH. Incorrect phrasing. Please rewrite the sentence.
  9. L227: Rephrase. e.g. The M2B operating region shifts from weak to strong inversion...
  10. L243: was simulated using...
  11. L272: simulation results across process, power supply voltage and temperature (PVT) variations.
  12. L277: the phrase "and the simulation results are shown in Fig.12. needs to be removed. Fig 12. is wrong in this context and Fig. 13, which is correct is mentioned in the next sentence.
  13. Table3. Comparison with Prior Arts.
